# Expansion Work Recovery of Hydrogen for a FC-Truck-Tentative Design of an Expansion Machine

Alfred Rufer

EPFL—Ecole Polytechnique Fédérale de Lausanne, CH1015 Lausanne, Switzerland; alfred.rufer@epfl.ch; Tel.: +41-79-244-09-84

**Abstract:** Hydrogen powered vehicles use high-pressure reservoirs from which the gas is transferred to the low-pressure fuel-cell via a classical pressure reduction valve. In these systems no expansion work is recovered and the question is addressed of the potential to increase global efficiency by using an expansion machine between the reservoir and the electrochemical reactor. This paper investigates the feasibility of such an expansion machine, and evaluates the mechanical constraints in terms of forces, torques and produced power by numeric simulation. It further evaluates the energetic contribution to the whole conversion chain from the hydrogen reservoir to the common electrical network on board. A low-energy contribution of the expansion system addresses the question of the real benefit of such an investment.

**Keywords:** hydrogen propulsion system; expansion work; expansion turbine; pressure reduction; energy efficiency





## 1. Introduction

In the context of the decarbonation actions and renouncement of fossil fuels, hydrogen is expected to play an important role in the future of transportation [1]. For all of the advantages of an electric propulsion system, the powering of vehicles with fuel-cells and hydrogen presents many advantages as a large range in comparison to battery EV's. One of the greatest advantages is the very short refuelling time [2]. Hydrogen on board is realized in the form of a high-pressure tank, but the operation of the fuel-cell converter only requires the pressure of 1–3 bar. For the reduction of pressure from the reservoir level to the fuel-cell, current vehicles use simple pressure reduction valves with cross section changes. In these systems, no work is recovered from the expansion process.

The potential of recovery of the expansion work of hydrogen for fuel-cell vehicles has been analyzed in [3]. The proposal is made for a replacement of the passive pressure reduction valve by an expansion engine followed by an electric generator in which output power is injected into the electric circuit of the vehicle for further use in propulsion for an expected increase of the normal range. This valuable principle addresses the question of available expansion machinery and their energetic performance. Figure 1 gives a schematic representation of a fuel-cell heavy vehicle with its main components. $H_2$ is flowing from the pressurized tank through the expansion machine before being converted into electric power in the fuel-cell. The expansion machine drives a dedicated generator from which the power is rectified by a power electronic converter and fed to the main electric circuit. In this diagram, a buffer battery is also shown as well as the propulsion machines. This paper is an attempt to design such an expansion machine and to evaluate its main parameters and constraints. Further, the average of the recovered power is calculated and compared to the power delivered by the fuel-cell. These calculations will be a base for the evaluation of the added value of the expansion equipment to the vehicle.

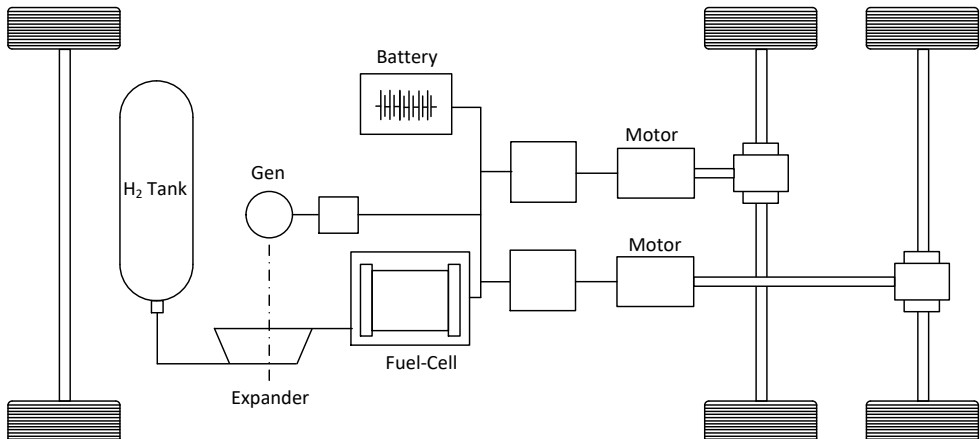

**Figure 1.** Schematic representation of a FCEV propulsion system.

The main issue of this study and development will be the improvement of the energy efficiency of systems using pressure reduction valves.

Expansion work recovery machines are dedicated first to the next generation of fuel-cell vehicles and can be developed by the future automotive industry. In a more general context of future energy storage facilities based on compressed hydrogen, they will improve the total energy efficiency of such systems. Applications to different fluids is also possible; for example, for compressed air driven machines and vehicles where pressure reduction valves are used today. Compressed Natural Gas (CNG) is another candidate for expansion machines.

## 2. Materials and Methods

Compressors for hydrogen-filling stations have been developed in recent years according to two dominant technologies. The leading technology in countries like Germany is the ionic compressor based on a liquid piston principle. Other systems use piston engines where the traditional pneumatic actuator has recently been replaced by a hydraulic cylinder for energetic efficiency reasons. A very recent development includes an original automatic seal exchange of the hydrogen high-pressure seal in order to reduce the revision downtime to a minimum [4]. Reversible compressors, namely, expansion machines adapted to the specific pressure range, are however not common industrial components and must be specially developed for the recovery function studied in this contribution.

### 2.1. A Dedicated Expansion Machine

2.1.1. Single-Sided Pistons with Crankshaft and Piston Rod

In this study, the considered machine is defined as a piston/cylinder arrangement where the principle of an intake-and-expand sequence will be implemented according to the principles described in [5–7]. Figure 2 shows the basic diagram of the expansion machine actuated by rotational equipment comprising a crankshaft and piston rods. The crankshaft is coupled to an electric motor via a toothed belt gear ratio. On both sides of the crankshaft, there are three cascaded cylinders for the intake, expansion and exhaust process.

The expansion of the gas is realized within two stages where the transfer of the fluid from $V_1$ to $V_2$ corresponds to a first stage of expansion, while the transfer from $V_2$ to $V_3$ corresponds to the second expansion stage. The three cascaded cylinders are characterized through their volumes ($V_1$, $V_2$, $V_3$), their surfaces ($S_1$, $S_2$, $S_3$) and the corresponding diameters of the pistons $D_1$, $D_2$, $D_3$. The pistons are operated in phase opposition with crank pins offset by 180 degrees. The cylinders of the machine are chosen as single-sided pistons for a unidirectional high-pressure solicitation of the sealings on one hand, and to avoid the piston-rod side seals on the other hand. Double acting cylinders would further have the disadvantage of dissymmetric left and right volumes with consequences on the

modulation of the global output torque. The six-cylinder architecture results from the principle of having two expansion stages where the second expansion uses the receiving (expansion) cylinder of the first stage as a source for the second expansion during the return stroke. If the use of an interstage heat exchange element is necessary, another architecture can be envisaged in order to avoid undesired dead volumes. The two expansion stages should be identical and each have a filling and an expansion cylinder. This means two cylinders per stage leading to a total of four. If the principle of reciprocating is maintained (180° shifted crankpins), the total number of cylinders becomes equal to eight.

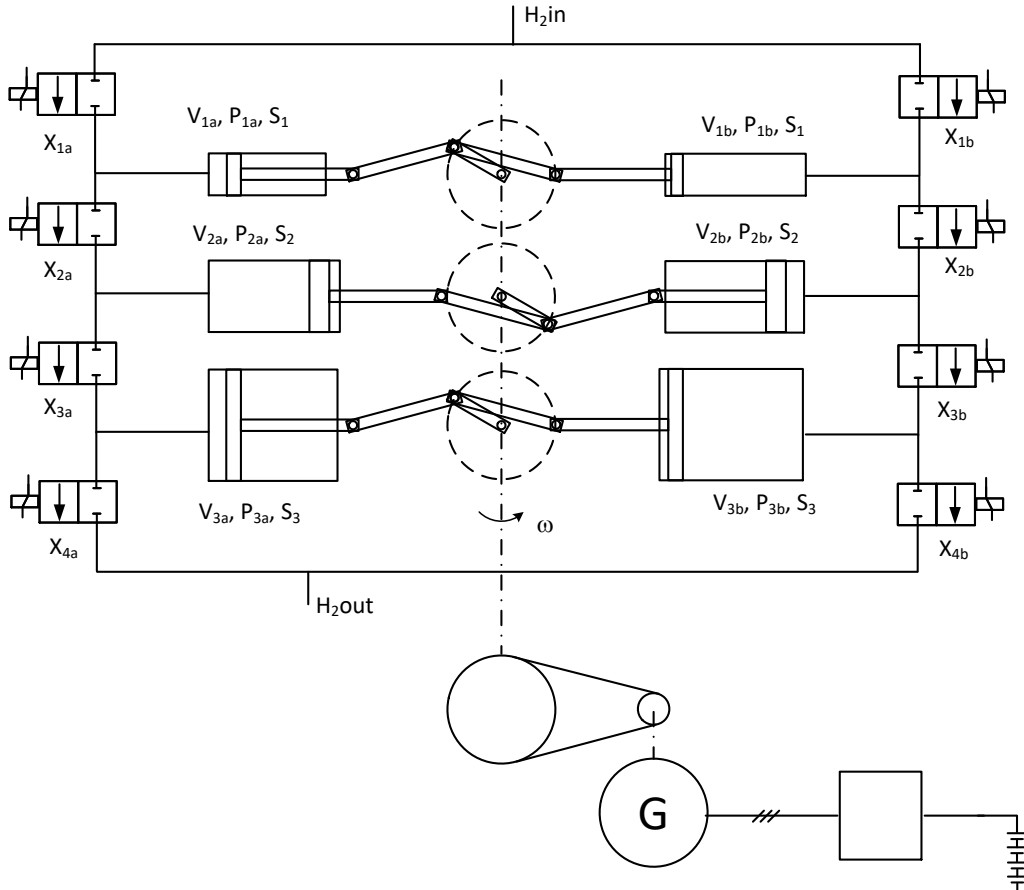

**Figure 2.** The six-cylinder two-stage expansion machine.

## 2.1.2. Basic Principle of the Expansion Machine

The basic principle of operation is described for the left-side arrangement of the three cylinders $V_{1a}$, $V_{2a}$ and $V_{3a}$. During the first half cycle, the small cylinder is filled with hydrogen at the constant high-pressure value. Then, after this filling, the gas is transferred to the larger cylinder $V_2$ during the return stroke. During this transfer, the gas occupies an evolutive volume starting from $V_1$ and reaching the value of $V_2$ at the end of the return stroke. The ratio of $V_1/V_2$ corresponds to the expansion factor. Simultaneously to the filling and first expansion within $V_1$ and $V_2$, a second expansion occurs in the next period with a further transfer of the gas from $V_2$ to $V_3$ before the expanded gas is exhausted from $V_3$ to the output during the following return stroke. The expansion factors are defined as $V_1/V_2$ for the first stage and $V_2/V_3$ for the second one. The pressures in the different volumes are represented in Figure 3 together with the control signals of the different valves.

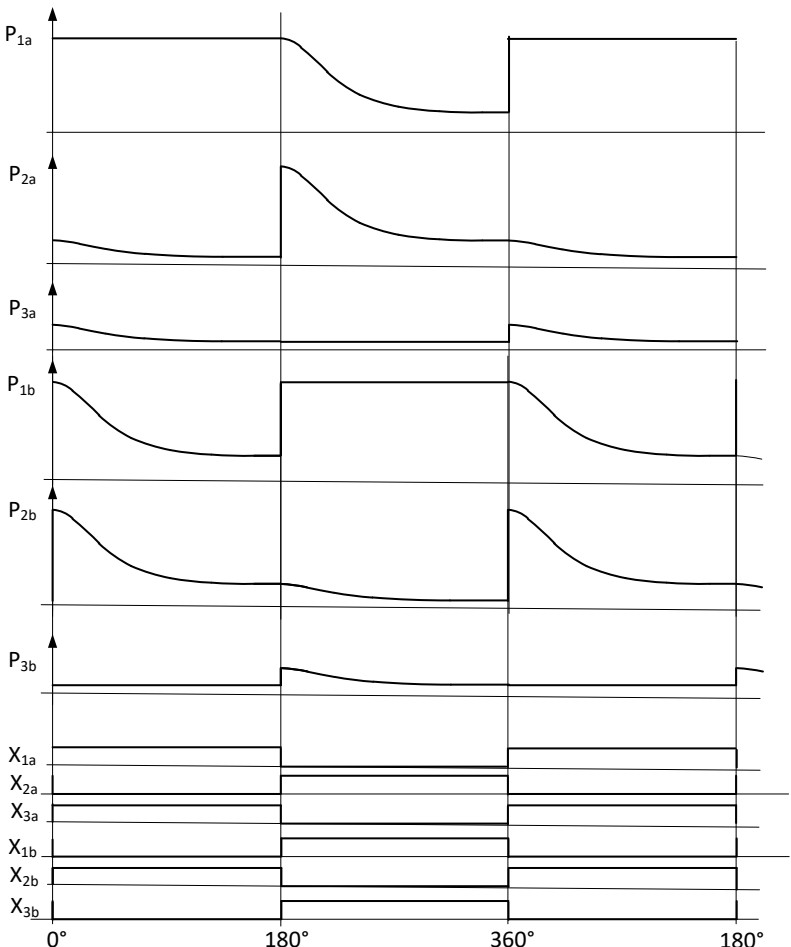

**Figure 3.** Pressures in the cylinders and control signals of the valves.

2.1.3. Estimation of the Needed Cylinder Volumes

The design of the expander is based on a maximum $H_2$ consumption of 4.8 g per second, which is related to a heavy truck [7,8]. During each half cycle, the first cylinders must intake and transfer half of that mass of gas, namely 2.4 g, when the expansion cycle is defined as 1 s. The choice for the cycle time of the expansion process will be explained in Section 3.1.2; it will also have a consequence of the piston's section and finally on the force exerted on it by the pressure. The mass of 2.4 g fills the first volumes after the intakes at a pressure of 350 bar or $350 \times 10^5$ N/m². The calculation of the volume of the first cylinders is based on the number of moles $n$ in the volume under a pressure of 350 bar.

$$n = \frac{m}{M} = \frac{2.4\text{g}}{2\text{g/mol}} = 1.2\text{mol,} \tag{1}$$

$M$ being the molar mass of hydrogen.

$$V_1 = \frac{n \cdot R \cdot T}{P} = \frac{1.2\text{mol} \cdot 8.31\text{J/mol} \times \text{K} \cdot 293\text{K}}{350 \cdot 10^5\text{N/m}^2} = 0.0000835\text{m}^3 \tag{2}$$

For the volume $V_2$, three times of the value of $V_1$ is considered. This comes from the choice of a total expansion factor equal to 9 for the cascade of the two stages. The expansion curve can be between an adiabatic or isothermal characteristic, and considering the worst

case under the lowest pressure in the reservoir, namely 50 bar, the output pressure of the expander in this case becomes equal to

$$P_3 = P_1 \left( \frac{V_1}{V_2} \cdot \frac{V_2}{V_3} \right)^{\gamma} = 50 \cdot 10^5 \cdot \left( \frac{1}{9} \right)^{1.4} = 2.3 \cdot 10^5 \text{N/m}^2 \tag{3}$$

$$V_2 = 3 \cdot V_1 = 0.00025 \text{m}^3 \tag{4}$$

For the third volume which is defined by the volume ratio of the second expansion stage, the same ratio is considered

$$V_3 = 3 \cdot V_2 = 0.00075 \text{m}^3 \tag{5}$$

Table 1 indicates the values of the pressures and temperatures after the two expansion stages in function of the input pressure, under adiabatic expansion. Table 2 indicates the same values but for the isothermal case. The calculated pressures correspond to

$$P_2 = P_1 \left( \frac{V_1}{V_2} \right)^{\gamma} \tag{6}$$

with $\gamma = 1.4$.

$$P_3 = P_2 \left( \frac{V_2}{V_3} \right)^{\gamma} \tag{7}$$

with $\gamma = 1.4$.

**Table 1.** Pressures and temperatures in function of the pressure of the reservoir $P_1$ in adiabatic conditions.

| Adiabatic Expansions | | | | | | | |
|---|---|---|---|---|---|---|---|
| $P_1$ | | $T_2$ (K) | $T_2$ (°C) | $P_2$ | $P_3$ | $T_3$ (K) | $T_3$ (°C) |
| 350 | 293 | 188.81 | −84.19 | 75.17 | 16.14 | 121.67 | −151.33 |
| 300 | 293 | 188.81 | −84.19 | 64.43 | 13.84 | 121.67 | −151.33 |
| 250 | 293 | 188.81 | −84.19 | 53.69 | 11.53 | 121.67 | −151.33 |
| 200 | 293 | 188.81 | −84.19 | 42.95 | 9.23 | 121.67 | −151.33 |
| 150 | 293 | 188.81 | −84.19 | 32.22 | 6.92 | 121.67 | −151.33 |
| 100 | 293 | 188.81 | −84.19 | 21.48 | 4.61 | 121.67 | −151.33 |
| 50 | 293 | 188.81 | −84.19 | 10.74 | 2.31 | 121.67 | −151.33 |

**Table 2.** Pressures and temperatures in function of the pressure in the reservoir $P_1$ in isothermal conditions.

| Isothermal Expansions | | | | | | | |
|---|---|---|---|---|---|---|---|
| $P_1$ | | $T_2$ (K) | $T_2$ (°C) | $P_2$ | $P_3$ | $T_3$ (K) | $T_3$ (°C) |
| 350 | 293 | 293 | 20.00 | 116.66 | 38.88 | 293 | 20.00 |
| 300 | 293 | 293 | 20.00 | 99.99 | 33.33 | 293 | 20.00 |
| 250 | 293 | 293 | 20.00 | 83.33 | 27.77 | 293 | 20.00 |
| 200 | 293 | 293 | 20.00 | 66.66 | 22.22 | 293 | 20.00 |
| 150 | 293 | 293 | 20.00 | 50.00 | 16.66 | 293 | 20.00 |
| 100 | 293 | 293 | 20.00 | 33.33 | 11.11 | 293 | 20.00 |
| 50 | 293 | 293 | 20.00 | 16.67 | 5.55 | 293 | 20.00 |

The values of the temperatures are given through

$$T_2 = \frac{T_1}{(V_2/V_1)^{\gamma-1}} \tag{8}$$

$$T_3 = \frac{T_2}{(V_3/V_2)^{\gamma-1}} \tag{9}$$

The calculated volumes of the three cylinders correspond to an input pressure of 350 bar. When the pressure in the reservoir decreases in dependency of the hydrogen consumption, a mass flow of 4.8 g/s can be maintained with the same volumes, but with an increase of the rotational speed of the machine.

2.1.4. Design of the Piston's Sections and Diameters

After having calculated the volumes of the cylinders, the sections and diameters of the pistons are calculated. For this calculation the radius of the crankshaft is first defined: r = 50 mm. Then, the stroke being 100 mm, the piston surfaces are calculated as

$$S_1 = \frac{V_1}{l} = \frac{0.0000835 \text{m}^3}{0.1\text{m}} = 0.000835 \text{m}^2 = 8.35 \text{cm}^2 \tag{10}$$

$$S_2 = \frac{V_2}{l} = \frac{0.00025 \text{m}^3}{0.1\text{m}} = 0.0025 \text{m}^2 = 25 \text{cm}^2 \tag{11}$$

$$S_3 = \frac{V_3}{l} = \frac{0.00075 \text{m}^3}{0.1\text{m}} = 0.0075 \text{m}^2 = 75 \text{cm}^2 \tag{12}$$

For the dimensions of the cylinders, the diameters $D_1$, $D_2$ and $D_3$ are calculated as

$$D_1 = \sqrt{\frac{4 \cdot S_1}{\pi}} = \sqrt{\frac{4 \cdot 8.35 \text{cm}^2}{\pi}} = 3.26 \text{cm} \tag{13}$$

$$D_2 = \sqrt{\frac{4 \cdot S_2}{\pi}} = \sqrt{\frac{4 \cdot 25 \text{cm}^2}{\pi}} = 5.6 \text{cm} \tag{14}$$

$$D_3 = \sqrt{\frac{4 \cdot S_3}{\pi}} = \sqrt{\frac{4 \cdot 75 \text{cm}^2}{\pi}} = 9.77 \text{cm} \tag{15}$$

With the values of the different pressures and the different sections of the pistons, the value of the forces exerted are calculated with

$$F = P \cdot S \tag{16}$$

Table 3 shows the maximum values of the forces of the three pistons. It can already be seen that the realization of such an expansion machine is a real challenge when the forces are in the range of tens of kilo Newton.

**Table 3.** Maximum forces of the pistons.

| $S_1$ [m²] | $P_1$max [N/m²] | $F_1$max [N] |
|---|---|---|
| $8.35 \times 10^{-4}$ | $3.50 \times 10^7$ | $2.92 \times 10^4$ |
| $S_2$ [m²] | $P_2$max [N/m²] | $F_2$max [N] |
| 0.0025 | $3.50 \times 10^7$ | $8.75 \times 10^4$ |
| $S_3$ [m²] | $P_3$max [N/m²] | $F_3$max [N] |
| $7.50 \times 10^{-3}$ | $7.52 \times 10^6$ | $5.64 \times 10^4$ |

## 3. Results

### 3.1. Modeling the Mechanical Constraints of the Special Machine

3.1.1. The Model of the Piston and Crankshaft

The next section gives the basic relations for a model of the piston and crankshaft assembly which will be used for the simulation of the mechanical and thermodynamic representation of the expansion machine [9]. In Figure 4, the different mechanical variables are defined.

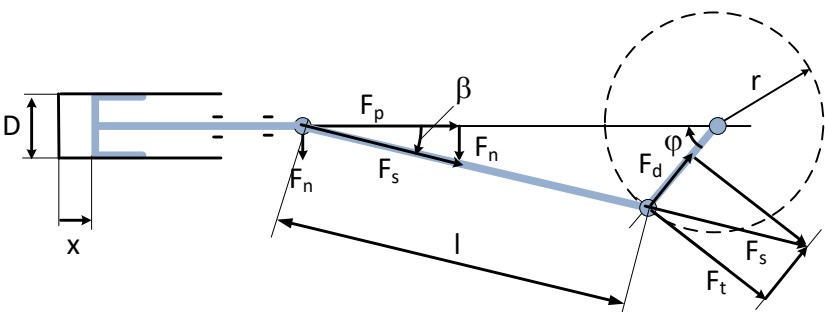

**Figure 4.** The variables of the piston and crankshaft assembly.

The horizontal displacement of the piston is determined through Equation (17)

$$x = r(1 - \cos \varphi) + \frac{\lambda}{2} r \sin^2 \varphi \tag{17}$$

In this relation, an additional parameter is used, namely the connecting rod ratio $\lambda$ which is defined as

$$\lambda = \frac{r}{l} \tag{18}$$

The piston's velocity is calculated according to Equation (19)

$$v = \omega \cdot r \cdot \sin \varphi (1 + \lambda \cos \varphi) \tag{19}$$

In this study, the torque developed by the motor is indirectly calculated through the power. The compression force exerted on the piston is given by the product of the pressure multiplied by the piston's front area A.

$$F_p = p \cdot A \tag{20}$$

The resulting mechanical power is defined by the product of the piston's force by its velocity.

$$Pow = F_p \cdot v \tag{21}$$

The torque is obtained by dividing the power by the angular velocity $\omega$

$$M_{mot} = Pow/\omega \tag{22}$$

With the other parameters and definition of the variables of Figure 4, the torque can be calculated without any consideration of the power and of the rotational speed [7]. As can be seen in the figure, the additional forces, such as the force which is parallel to the connecting rod $F_s$, the tangential force $F_t$ and the reaction force $F_n$, are also defined. The torque on the crankshaft is given by Equation (23).

$$M_{mot} = F_t \cdot r \tag{23}$$

The tangential force $F_t$ is given by (24)

$$F_t = F_s \cos(\pi/2 - (\beta + \varphi)) \tag{24}$$

The force transmitted through the connecting rod $F_s$ is calculated with the help of the piston force $F_p$ and the angle beta (26).

$$F_s = \frac{F_p}{\cos \beta} \tag{25}$$

$$\beta = arc \sin\left(\frac{r}{l} \cdot \sin \varphi\right) \tag{26}$$

In Figure 4, the reaction force $F_n$ which is perpendicular to the piston's movement is given as per (27).

$$F_n = F_s \sin \beta = \frac{F_p}{\cos \beta} \sin \beta \tag{27}$$

For this system using a crankshaft and a piston rod, a specific linear guiding element is needed which can hold the perpendicular reaction force at the bottom side of the piston. In this architecture, the piston rod cannot move inside the compression cylinder like in a normal piston assembly of a classical internal combustion engine. The consequence is that the total length of the compressor (in the direction of the piston's displacement) is significantly increased.

### 3.1.2. Simulation of a Slow Expansion Machine

Related to the very low temperature levels reached after the two expansions, and in reference to the operation conditions of high-pressure gas boosters from the literature, a machine with slow rotational speed is chosen. The slow rotation and slow motion of the pistons will allow a better transfer of the heat from the cylinder walls to the expanded gas and produces a displacement of the p-v expansion curve from an adiabatic characteristic towards an isothermal one.

The rotational speed is supposed to be constant and equal to 6.28 rad/s. Figure 5 shows the evolution of the pistons Nr. 1a and Nr.2a (according to the definition of Figure 2) and Figure 6 shows the evolution of the volumes $V_{1a}$, $V_{2a}$ and $V_{3a}$. The position of the third piston (Nr 3a) is identic to the position of piston Nr. 1a.

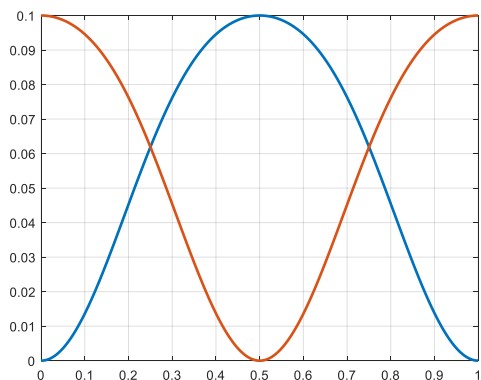

x-axis time [s], y-axis positions [m]

**Figure 5.** Position of the first (blue) and second (red) pistons.

In dependency of the variation of the volumes and taking into account the transfer of the gas from one cylinder to the other, the pressures in the different cylinders are calculated (Equations (6) and (7)). Then, the forces exerted on the pistons are calculated (Equation (20)). The waveforms of the pressures and of the respective forces are similar, but the values and units are totally different. In Figures 7–12, the pressures and forces of the pistons a and b of cylinders 1, 2 and 3 are represented.

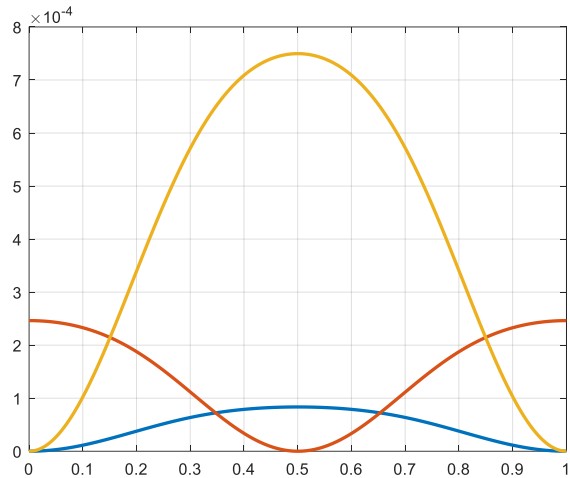

x-axis: time [s]; y-axis: $V_{1a}$, $V_{2a}$, $V_{3a}$ [m³]

**Figure 6.** Volumes of the cylinders.

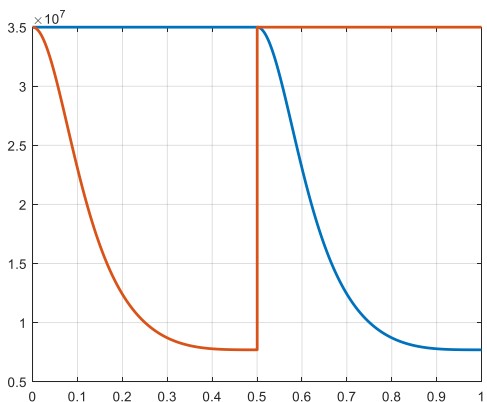

x-axis: time [s]; y-axis: pressure 1a (red), 1b (blue)

**Figure 7.** Pressure 1a and 1b [N/m²].

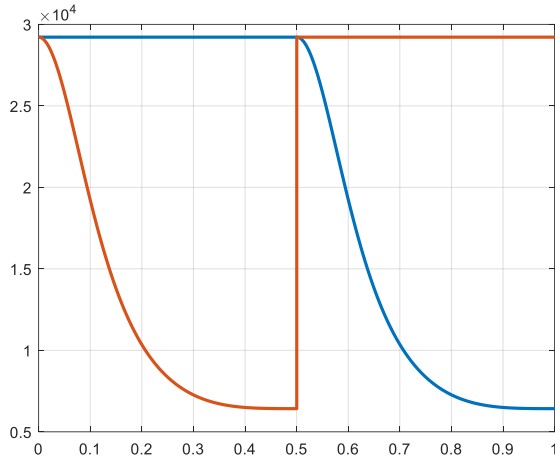

x-axis: time [s]; y-axis: Forces 1a (red) and 1b (blue)

**Figure 8.** Force 1a and 1b [N].

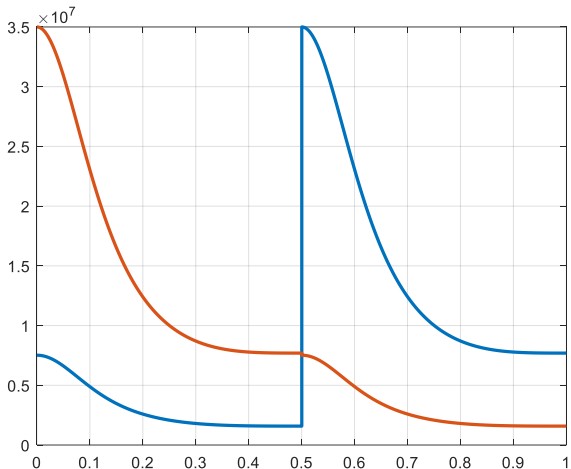

x-axis: time [s]; y-axis Pressures 2a (red), 1b (blue)

**Figure 9.** Pressure 2a and 2b [N/m$^2$].

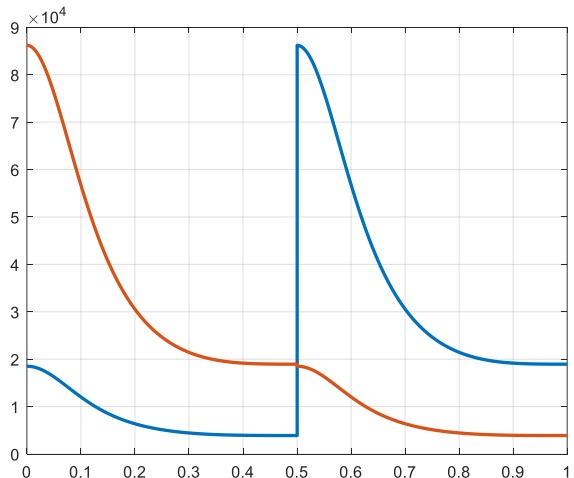

x-axis: time [s]; y-axis: Forces 2a (red) and 2b (blue)

**Figure 10.** Force 2a and 2b [N].

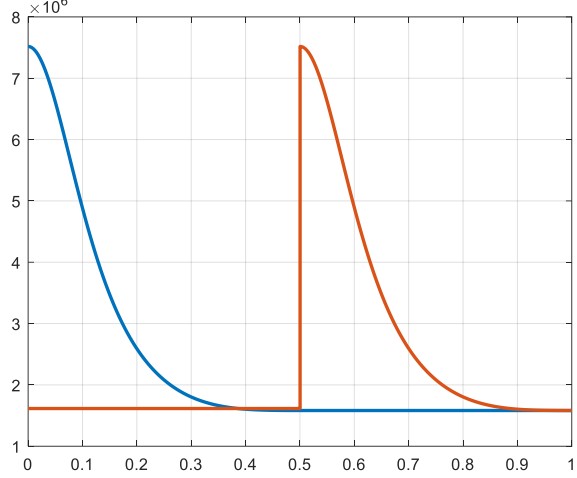

x-axis: time [s]; y-axis: Pressures 3a (blue) and 3b (red)

**Figure 11.** Pressure 3a and 3b [N/m$^2$].

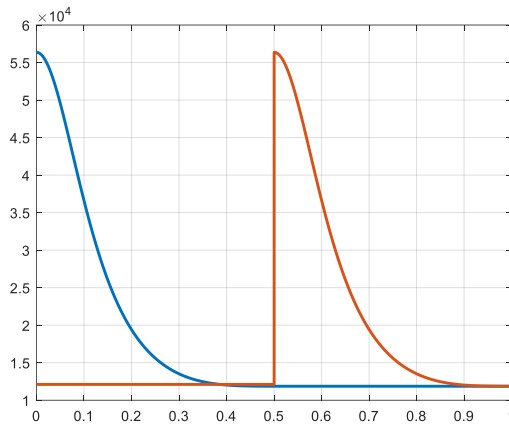

x-axis: time [s]; y-axis:Forces 3a (blue) and 3b (red)

**Figure 12.** Force 3a and 3b [N].

Further, multiplying the forces by the velocity of the pistons, the power produced is calculated (Equation (21)). By dividing by the rotational angular speed the torque produced is also calculated (Equation (22)). In Figures 13–18, the power and torque of each piston and at each side (a and b) is represented. The maximum values of the torques produced by the pistons are given in Table 4.

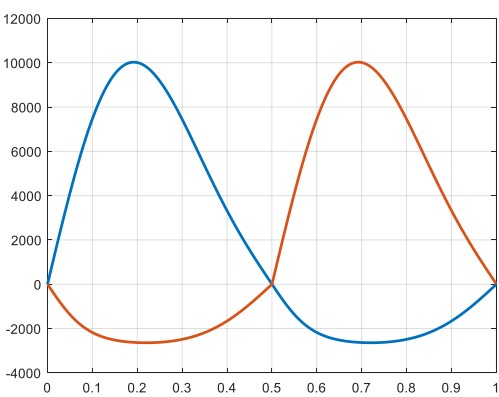

x-axis: time [s]; y-axis Power 1a (blue) and 1b (red)

**Figure 13.** Power 1a and 1b [W].

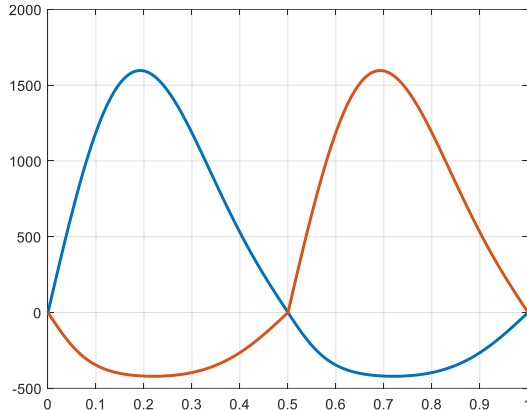

x-axis: time [s]; y-axis: Torques 1a (blue) and 1b (red)

**Figure 14.** Torque 1a and 1b [Nm].

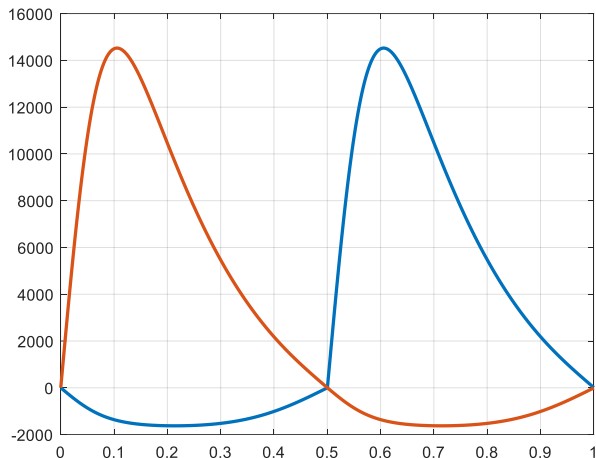

x-axis: time [s]; y-axis: Power 2a (red) and 2b (blue)

**Figure 15.** Power 2a and 2b [W].

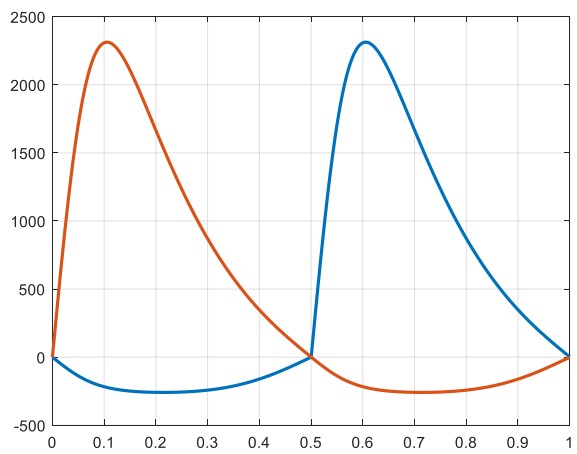

x-axis: time [s]; y-axix: Torque 2a (red) and 2b (blue)

**Figure 16.** Torque 2a and 2b [Nm].

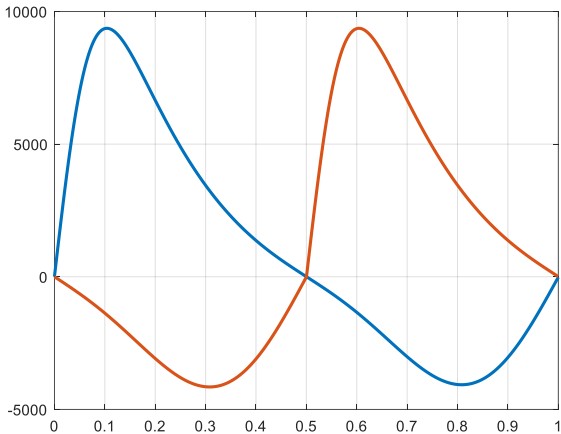

x-axis: time [s]; y-axis: Power 3a (blue) and 3b (red)

**Figure 17.** Power 3 and 3b [W].

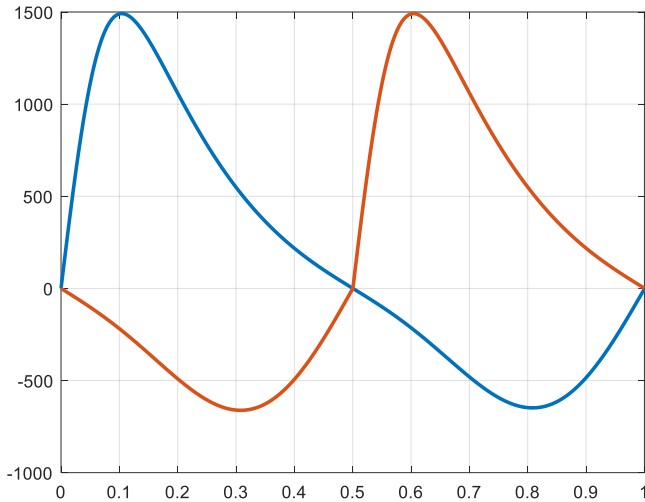

x-axis: time [s; y-axis: Torque 3a (blue) and 3b (red)]

**Figure 18.** Torque 3a and 3b [Nm].

**Table 4.** Maximum values of the torques.

| Maximal Torque | | |
|---|---|---|
| $M_1$ [Nm] | $M_2$ [Nm] | $M_3$ [Nm] |
| $1.60 \times 10^3$ | $2.31 \times 10^3$ | $1.49 \times 10^3$ |

Finally, the sum of the three powers giving the total power delivered by the expander is represented in Figure 19. The total torque produced is represented in Figure 20. The curves of Figures 19 and 20 correspond to the performance of the expander for adiabatic expansion conditions.

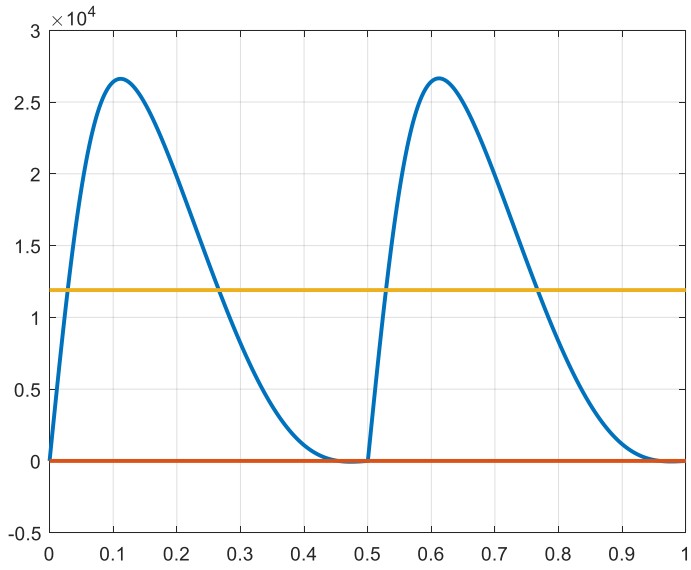

x-axis: time [s]; y-axis: Total power (blue) and average (red)

**Figure 19.** Total power and average (adiabatic) [W].

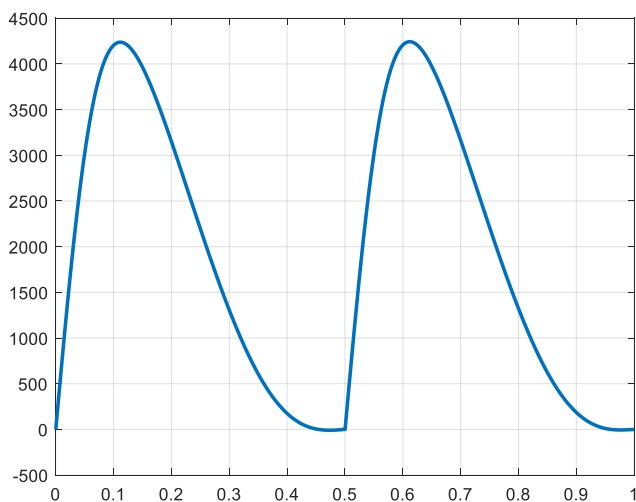

x-axis: time [s]; y-axis: Total torque

**Figure 20.** Total torque (adiabatic) [Nm].

### 3.1.3. Isothermal Expansion

The expansions can be assumed to be realized according to an isothermal characteristic. Even if this cannot be achieved totally in a real system, the produced power and torque are slightly higher than in adiabatic conditions. The following simulations are realized for an evaluation of such a best case and show the most important system variables in isothermal conditions. To show the pressure variations during the expansions, the pressure in the second pistons is represented (Figure 21). Effectively, the pressure in the second cylinder also shows the variation of the pressure during the transfer from the first cylinder into the second during the first half-cycle, and the variation during the transfer from this second cylinder into the third one during the return stroke.

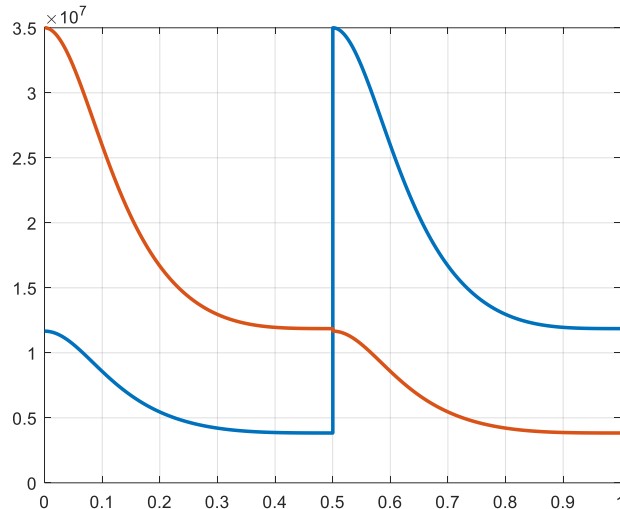

x-axis: time [s]; y-axis: Pressure 2a (red) and 2b (blue)

**Figure 21.** Pressure 2a and 2b (isothermal) [N/m$^2$].

Figure 22 shows the instant value and the average of the produced total power in isothermal conditions. Figure 23 shows the corresponding torque.

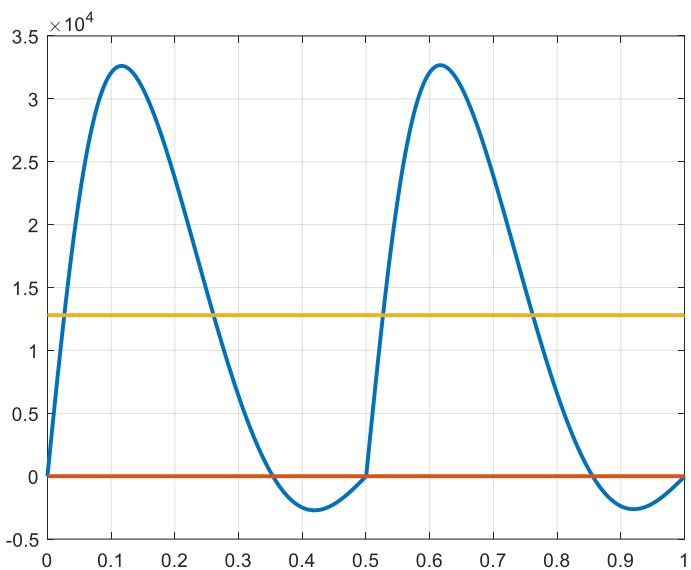

x-axis: time [s]; y-axis: Total power (isothermal)

**Figure 22.** Total power and average (isothermal) [W].

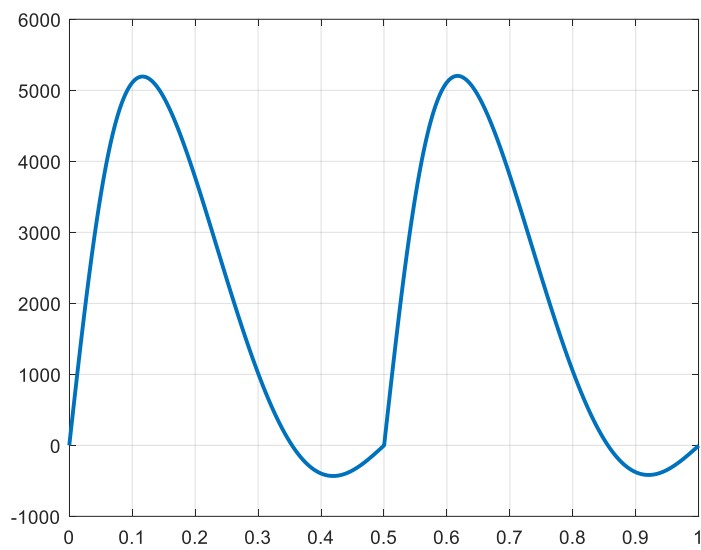

x-axis: time [s]; y-axis: Total torque (isothermal)

**Figure 23.** Total torque (isothermal) [Nm].

### 3.2. A Faster Machine with Reduced Size and Constraints

The first design of the new expansion machine shows that, due to the high pressure of the gas, a low-speed machine leads to extreme mechanical constraints in terms of forces and torques. Consequently, a faster-running machine is designed and analyzed.

### 3.2.1. Dimensions of the Faster Machine

If the volume of the first cylinder is $V_1 = 0.0000835$ m$^3$ and corresponds to a hydrogen flow of 2.4 g/s when the period of the cycle is 1 s, the volume of the cylinder becomes

$$V_{1f} = V_1/10 = 0.00000835 \text{m}^3$$

when the period is chosen as 0.1 s.

For the volume $V_2$, three times of the value of $V_1$ is considered

$$V_{2f} = 3 \cdot V_{1f} = 0.000025\text{m}^3 \tag{28}$$

For the third volume which defines the volumetric ratio of the second expansion stage, the same ratio is considered

$$V_{3f} = 3 \cdot V_{2f} = 0.000075\text{m}^3 \tag{29}$$

In order to keep the diameter of the small piston at a reasonable value, the stroke of the pistons is reduced to 0.05 m. Then the section of the small piston becomes

$$S_{1f} = \frac{V_{1f}}{l} = \frac{0.00008835\text{m}^3}{0.05\text{m}} = 0.000167\text{m}^2 = 1.67\text{cm}^2 \tag{30}$$

$$S_{2f} = \frac{V_{2f}}{l} = \frac{0.000025\text{m}^3}{0.05\text{m}} = 0.0005\text{m}^2 = 5\text{cm}^2 \tag{31}$$

$$S_{3f} = \frac{V_{3f}}{l} = \frac{0.000075\text{m}^3}{0.05\text{m}} = 0.0015\text{m}^2 = 15\text{cm}^2 \tag{32}$$

For the dimensions of the cylinders, the diameters $D_1$, $D_2$ and $D_3$ are calculated as

$$D_{1f} = \sqrt{\frac{4 \cdot S_{1f}}{\pi}} = \sqrt{\frac{4 \cdot 1.67\text{cm}^2}{\pi}} = 1.46\text{cm} \tag{33}$$

$$D_{2f} = \sqrt{\frac{4 \cdot S_{2f}}{\pi}} = \sqrt{\frac{4 \cdot 5\text{cm}^2}{\pi}} = 2.52\text{cm} \tag{34}$$

$$D_{3f} = \sqrt{\frac{4 \cdot S_{3f}}{\pi}} = \sqrt{\frac{4 \cdot 15\text{cm}^2}{\pi}} = 4.37\text{cm} \tag{35}$$

With the faster machine, the expansion factors are identical to the case of the previous machine. When the piston's surfaces are different, the maximum of the forces is also different. Table 5 gives the maximum values of the produced forces in the case of the fast machine.

**Table 5.** Maximal pressures and forces of the faster machine (adiabatic expansions).

| $S_{1f}$ [m$^2$] | $P_{1f}$max [N/m$^2$] | $F_{1f}$ [N] |
|---|---|---|
| $1.67 \times 10^{-4}$ | $3.50 \times 10^7$ | $5.85 \times 10^3$ |
| $S_{2f}$ [m$^2$] | $P_{2f}$max [N/m$^2$] | $F_{2f}$max [N] |
| 0.0005 | $3.50 \times 10^7$ | $1.75 \times 10^4$ |
| $S_{3f}$ [m$^2$] | $P_{3f}$max [N/m$^2$] | $F_{3f}$max [N] |
| $1.50 \times 10^{-3}$ | $7.52 \times 10^6$ | $1.13 \times 10^4$ |

### 3.2.2. Simulation of the Faster Machine

In a similar way as was made for the slow machine, the different variables are simulated. Figure 24 show the pressure in the cylinders 2a and 2b. The corresponding forces are shown in Figure 25. The waveforms are similar to the waveforms of the slow machine, but with values related to the new surfaces of the pistons. The values on the horizontal axis (time) are of course related to the new period of 0.1 s.

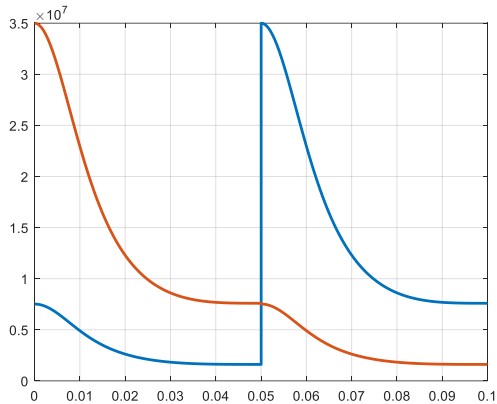

x-axis: time [s]; y-axis: Pressure 2a (red) and 2b (blue)

**Figure 24.** Pressure 2a and 2b [N/m$^2$].

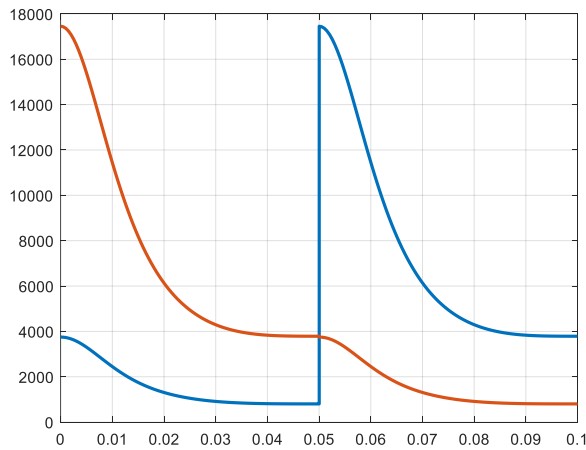

x-axis: time [s]; y-axis: Force 2a (red) and 2b (blue)

**Figure 25.** Force 2a and 2b [N].

In Figure 26, the total power and its average value are represented. The torque is represented in Figure 27. These values are strongly reduced in comparison with the values obtained for the slow machine.

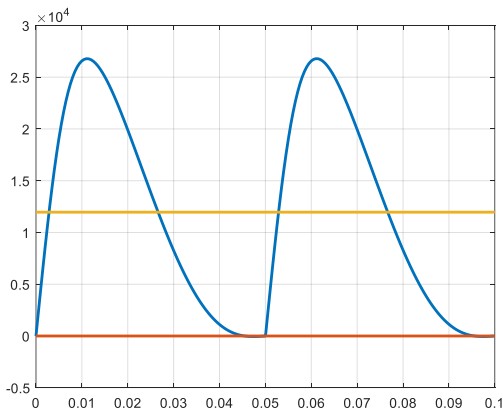

x-axis: time [s]; y-axis: Total power (blue) and verage (red)

**Figure 26.** Total power and average [W].

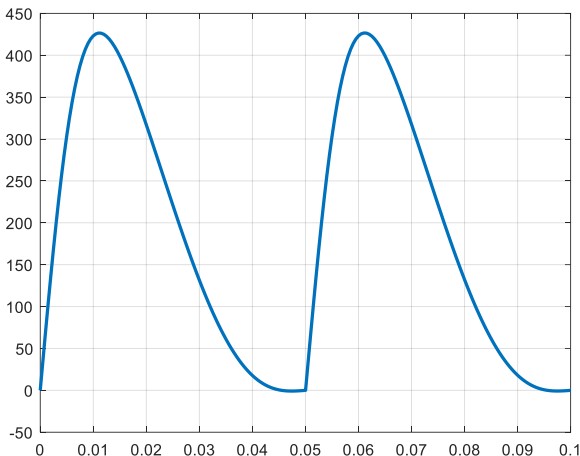

**Figure 27.** Total torque [Nm]; x-axis: time [s].

### 3.3. The Temperature in the Cylinders

The decrease of the temperature during the expansion is calculated through

$$T_2 = \frac{T_1}{(V_2/V_1)^{\gamma-1}} \tag{36}$$

For the system described by the schematic diagram of Figure 2, there are two successive expansions leading to two successive decreases of temperature. The variation of temperature illustrating the two successive expansions can be observed in the second cylinder $V_2$. During the first half period, the cylinder $V_2$ is participating in a first expansion due to the transfer of hydrogen from cylinder $V_1$ to cylinder $V_2$. During the second half period, the same cylinder V$_2$ is implicated in the second expansion where the gas is transferred from $V_2$ to $V_3$. The temperature curve of the double adiabatic expansion process is represented in Figure 28.

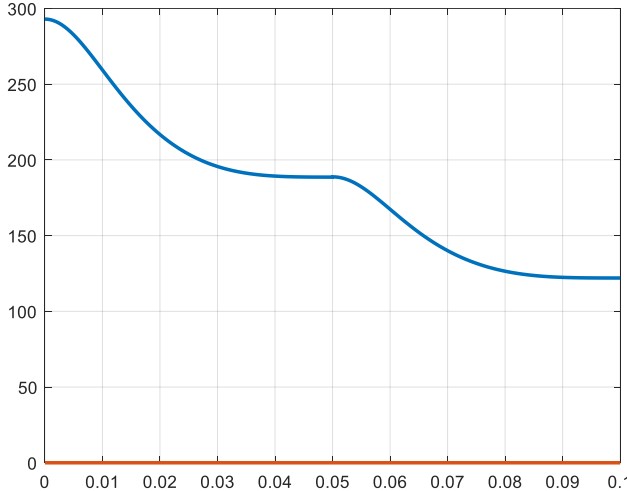

x-axis: time [s]; y-axis: Temperature cylinder 2

**Figure 28.** Decrease of the temperature in cylinder 2 (adiabatic expansion) [K].

In Table 1, the final temperatures of the adiabatic expansions have been indicated ($T_2$ and $T_3$), the initial value of the hydrogen gas being 293 K before the expansions. The curve in Figure 28 shows that the temperature in cylinder 2 reaches asymptotically the same values of Table 1, namely $T_2$ = 188 K and $T_3$ = 121 K.

### 3.4. Energetic Considerations

The power capacity of an expansion system has been estimated in the sections dedicated to the machinery. From Figures 19 and 26, the curves of the power produced by the two-stage reciprocating expanders show that the values are identic and that the output power is independent from the chosen nominal value of the rotational speed of the machine. By neglecting the friction of the piston seals and other parasitic losses, the amount of produced power is only depending on the flowrate of hydrogen to be expanded from the input pressure level to the output of the expander. Of course, the power will depend also on the quality of the expansion formulated through the polytropic factor. An isothermal expansion produces a higher power than an adiabatic one (Figure 22). The average value of the produced power in adiabatic conditions is 11.9 kW and can be converted into an electric power $P_{Gen}$ by the generator driven by the expansion machine (Figure 1). The efficiency of the generator (Permanent Magnet Synchronous Generator) is evaluated as 93%. The value of the power produced and converted in the recovery chain will now be compared to the electric power $P_{FC}$ produced by the fuel-cell which converts the expanded amount of hydrogen. The power ratio of the recovered expansion power to the useful power $P_{FC}$ is then defined as

$$R_{recov} = \frac{P_{Gen}}{P_{FC}} \tag{37}$$

where $P_{FC}$ is calculated from the $H_2$ flowrate of 4.8 g/s, with a heat value of $H_2$ of 33 kW/kg and an efficiency of the fuel-cell of 60%

$$P_{FC} = 4.8\text{g/s} \cdot 33\text{Wh/g} \cdot 3600\text{s/h} \cdot 0.6 = 342\text{kW} \tag{38}$$

The recovery factor $R_{recov}$ becomes

$$R_{recov} = \frac{P_{Gen}}{P_{FC}} = \frac{11.9\text{kW} \cdot 0.93}{342\text{kW}} = 0.032 \quad or \quad 3.2\% \tag{39}$$

The expansion power in isothermal conditions has been evaluated (see Figure 22); the recovery factor in this case becomes

$$R_{recov} = \frac{P_{Gen}}{P_{FC}} = \frac{12.8\text{kW} \cdot 0.93}{342\text{kW}} = 0.0348 \quad or \quad 3.48\% \tag{40}$$

This low value addresses the question of the motivation or the economic aspect of using expansion recovery systems in fuel-cell vehicles. An evaluation of the dimensions and weight of the components of the expansion system is necessary.

## 4. Discussion

The realization of an expansion machine for the recovery of the expansion work of hydrogen in a fuel-cell vehicle represent an enormous challenge for designers at the level of strong mechanical constraints such as forces and torques. A two-stage six-cylinder expander has been studied, where the rotational speed was kept at a low level due to the thermodynamic characteristic and to keep the expansion away from an adiabatic performance. In dependency of the variation of the pressure in the vehicle's reservoir, the rotational speed can be increased when the gas mass flowrate demand corresponds to the nominal value of 4.8 g/s for a heavy truck. Starting from the nominal reservoir pressure of 350 bar, the mechanical constraints as the typical high torque of thousands of Nm addresses the question of its feasibility or of the size and weight of the equipment. A 10-times faster machine was also simulated, bringing a strong reduction of the constraints.

Adiabatic and isothermal expansions have been simulated, showing a limited performance increase between adiabatic and isothermal, even if the advantages of the nearly isothermal system would be useful for the thermal solicitation of cylinder and piston seals.

From the point of view of the amount of power produced by the gas expansion system in comparison to the power delivered from the chemical content of the gas through the fuel-cell, a limited benefit of only a few percent has been calculated.

## 5. Conclusions

A tentative design of a hydrogen expansion work recovery system has been realized where the pressures, forces, torques and power contributions of the two stages of a six-cylinder machine have been calculated by simulation. The huge mechanical constraints in a slow rotational speed machine could be strongly reduced by a second design of a faster running machine.

The limited power contribution of the expansion system addresses the technical and economic advantages of such a development. Implementing the studied expansion machine with the coupled generator and converter would increase the total weight of the vehicle, being a penalty for its performance in terms of additional fuel consumption, and corresponding range.

**Funding:** This research received no external funding.

**Data Availability Statement:** Not applicable.

**Conflicts of Interest:** The author declares no conflict of interest.

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
