# Peer review of "Expansion Work Recovery of Hydrogen for a FC-Truck-Tentative Design of an Expansion Machine"

_inventions, doi:10.3390/inventions8040089_

Round 1

Reviewer 1 Report

The manuscript entitled “inventions-2481980” dealing with simulation has been reviewed. The paper has been nicely written but needs some improvement. Please follow my comments.

1.     Authors are encouraged to state what industry applicartion of this work.

2.     Figure 2 shows six cylinder system. Is this work suitable for another arrangement for instance 4 or 8 cylinder?

3.     What is the main issue that will be addressed by this research? This needs to be highlighted.

4.     Can you provide some quanitative info for the presented equations? For example Equation 27.

5.     What is the future direction of this work?

6.     Please proofread the paper.

7.     Simulation has different application in science and industry. Please read and add the following reference in the application of simulation in manufacturing and highlight the application of your research. “Benchmark models for conduction and keyhole modes in laser-based powder bed fusion of Inconel 718”.

Needs some improvements.

Author Response

Answers to Reviewer 1

  1. Industry applications : An additional paragraph has been introduced in the text (lines 46-51)
  2. 4 or 8 cyinders: Additional text introduced (lines 86-93)
  3. Additiolal text has been introduced
  4. Quantitative info to the presented equations: In the example of rel. 27, there is only a proportionality fctor between two forces which is formulated. The factor is dimension less. All variables are defined with their corresponding units, Nm for torques, N for forces, N/m2 for pressure etc.
  5. The future direction of the work would be to go experimental, and to verify the thermal behaviour of the expansion machine, (need and performance of stage – and inter-sage heaters, limitation of the temperature decrease. Excess heat from the fuel-cell would be a first attempt.
  6. Paper proofread

7. The recommended reference is evaluated to be far away from the proposed paper. Simulation is a common method for developing innovative machines and processes, and especially for representing the evolution of internal variables. The author is of the meaning that there is no need for the argumentation of using simulation.

Reviewer 2 Report

In this paper, the authors have explored the potential of utilizing an expansion machine between the reservoir and the electrochemical reactor to enhance overall efficiency. Through numerical simulations, the authors assess mechanical constraints, such as forces, torques, and power generation. They also evaluate the energetic contribution of the entire conversion chain, from the hydrogen reservoir to the onboard electrical network. Overall, this paper is interesting and merits its publication in Inventions. The reviewer recommends it to be accepted if the following comments could be properly addressed.

1. The authors may need to provide comparative results to demonstrate the performance difference between the system with and without the expansion machine.

2. It would be valuable if the authors could discuss the limitations associated with implementing their design in practical scenarios.

Moderate editing of English language required.

Author Response

Answers to Reviewer 2

  1. Comparative results to demonstrate the performance difference between a system with and without an expansion machine has been given through the indicated “recovery factor” where the power (or energy) that is recovered and added to the output of the fuel-cell is evaluated in percent (3.2 and 3.48% increase of the power in comparison to a common vehicle driven by fuel-cell where a pressure reduction valve is used. An additional calculation of the ratio of the theoretical expansion energy versus the chemical energy content has been done by the author. But this calculation is intentionally not include in the paper because it would be redundant.

2. The limitations associated with implementing the new machine is shortly discussed  (added text in the file)
